# Xyloglucan, Hibiscus and Propolis in the Management of Uncomplicated Lower Urinary Tract Infections: A Systematic Review and Meta-Analysis

**DOI:** 10.3390/antibiotics11010014

**Published:** 2021-12-23

**Authors:** Tommaso Cai, Umberto Anceschi, Irene Tamanini, Serena Migno, Michele Rizzo, Giovanni Liguori, Alejandro Garcia-Larrosa, Alessandro Palmieri, Paolo Verze, Vincenzo Mirone, Truls E. Bjerklund Johansen

**Affiliations:** 1Department of Urology, Santa Chiara Regional Hospital, 38123 Trento, Italy; irene.tamanini@apss.tn.it; 2Institute of Clinical Medicine, University of Oslo, 0010 Oslo, Norway; t.e.b.johansen@medisin.uio.no; 3Department of Urology, IRCCS Regina Elena National Cancer Institute, 00100 Rome, Italy; umberto.anceschi@gmail.com; 4Department of Gynaecology and Obstetrics, Santa Chiara Regional Hospital, 38123 Trento, Italy; serena.migno@apss.tn.it; 5Department of Urology, University of Trieste, 34121 Trieste, Italy; mik.rizzo@gmail.com (M.R.); gioliguori33@gmail.com (G.L.); 6Department of Urology, Hospital del Mar, 08001 Barcelona, Spain; garcialarrosa@yahoo.es; 7Department of Urology, University of Naples, Federico II, 80100 Naples, Italy; info@alessandropalmieri.it (A.P.); mirone@unina.it (V.M.); 8Department of Urology, University of Salerno, 84121 Salerno, Italy; pverze@unisa.it; 9Department of Urology, Oslo University Hospital, 0010 Oslo, Norway; 10Institute of Clinical Medicine, University of Aarhus, 8000 Aarhus, Denmark

**Keywords:** antibiotic resistance, urinary tract infection, uropathogens, xyloglucan, hibiscus, propolis, review, mucoprotection

## Abstract

Background: In the era of antibiotic resistance, an antibiotic-sparing approach presents an interesting alternative treatment of uncomplicated cystitis in women. Our aim is to perform a systematic review and meta-analysis to compare the effectiveness and safety profile of a medical device containing xyloglucan, hibiscus and propolis (XHP) in women with uncomplicated cystitis. Methods: Relevant databases were searched using methods recommended by the Preferred Reporting Items for Systematic Reviews and Meta-Analysis guidelines. The primary endpoint was clinical or microbiological success, defined as the complete (cure) and/or non-complete (improvement) resolution of symptoms at the end of treatment, or microbiological resolutions. Results: After screening 21 articles, three studies were included, recruiting a total of 178 patients. All three studies used placebo as comparator. A statistically significant difference was found in terms of clinical or microbiological resolution between the medical device and the comparator (3 RCTs, 178 patients, OR: 0.13; 95% CI: 0.05–0.33; *p* < 0.0001). No clinically significant adverse effects have been reported. Conclusion: A medical device containing xyloglucan, hibiscus and propolis is superior to comparator regimens in terms of clinical effectiveness in adult women with microbiologically confirmed or clinical suspicion of uncomplicated cystitis and is associated with a high patient compliance.

## 1. Introduction

The prevalence of recurrent uncomplicated urinary tract infections (rUTIs) in women accounts for about 27% and the impact of those on patients’ quality of life is high [1,2]. In particular, suffering from recurrent urinary tract infection has a detrimental influence on patient quality of life, in terms of the level of stress and anxiety [2]. In addition, its management is still under discussion and there is not a consensus [3,4]. The need for an antibiotics-sparing approach, due to the worldwide emergency of antibiotic resistance, and the recent interest in the role of non-antibiotic oral supplements increased the interest among physicians and researchers in the use of medical devices containing phytotherapy and nutraceuticals compounds. On the basis of this considerations, here we focused our attention on a medical device containing xyloglucan, hibiscus and propolis (XHP) in the management of women affected by rUTIs. Recently Cai et al. demonstrated in a prospective clinical trial the clinical and microbiological efficacy of XHP in improving quality of life in women affected by rUTIs, reducing symptomatic episode and the use of antibiotics [5]. This study has been based on the encouraging results from previous published RCTs [6,7,8]. In this sense, the use of XHP seems an interesting tool for use in everyday clinical practice for improving antimicrobial stewardship. Considering this background information and considering the lack of a general consensus about the management of rUTIs, we aimed to compare the effectiveness and safety profile of XHP in women with uncomplicated cystitis, by performing a systematic review and meta-analysis of relevant randomized controlled trials.

### Research Questions

We put forth two research queries:Is XHP able to obtain significant pre-clinical data in order to justify its clinical use in the management of patients affected by uncomplicated cystitis?Is XHP able to obtain a clinical and/or microbiological cure in women affected by uncomplicated cystitis?

## 2. Results

Our research produced 21 potentially relevant articles. After the screening, three pre-clinical studies and four clinical trials were considered. All the papers were published after 2015. The detailed selection process of the included trials has been displayed in Figure 1.

### 2.1. Pre-Clinical and Non-Randomized Clinical Studies

Three pre-clinical studies were included: one in vitro study and two animal model studies. One prospective non-randomized clinical study has been included in the systematic review but not in the meta-analysis. Table 1 shows characteristics of all of the pre-clinical and non-randomized clinical studies.

### 2.2. RCTs

#### 2.2.1. Evidence Synthesis, Study Characteristics and Quality Assessment Results

Three RCTs were considered eligible to be included in the meta-analysis, recruiting a total of 178 patients. One study was performed with a combination of xyloglucan, hibiscus and propolis, and two studies were performed with a combination of an equivalent mucoprotectant (reticulated gelatin), hibiscus and propolis. The Jadad score was 4 for two papers and 5 for one paper. One paper also includes an account of all patients, and for this reason, the trialists assigned the score of 5. The risk of publication bias and small-study effects has been evaluated with funnel plot analysis. The characteristics of all RCTs included in the systematic review and meta-analysis are described in Table 2.

#### 2.2.2. Characteristics of Studies Included in the Meta-Analysis

In two studies, the medical device is used in the management of acute uncomplicated cystitis treatment as adjuvant therapy in association with an antimicrobial agent. In one study, the medical device is used in the management of acute uncomplicated cystitis treatment alone. In two studies, the medical device is used in association with an antimicrobial agent and used for 2 months. On the other hand, in one study, the medical device is used only for 5 days in acute setting. The follow-up period ranges from 11 days to 2 months.

#### 2.2.3. Clinical Cure

We considered all three RCTs for the evaluation of a clinical cure. A statistically significant difference has been found in terms of clinical resolution between the medical device and the comparator (3 RCTs, 178 patients, OR: 0.13; 95% CI: 0.05–0.33; *p* < 0.0001). Figure 2 shows the clinical success (improvement/cure) in women with cystitis who were treated with the medical device compared to other antibiotic agents. Funnel plots were generated to analyze publication bias and small-study effects. The funnel plots analysis did not suggest the exclusion of any study (Figure 3).

#### 2.2.4. Safety Outcomes

No significant difference has been found in terms of adverse effects (3 RCTs, 178 patients, OR: 0.14; 95% CI: 0.03–0.67; *p* = 0.001) (Figure 4). The most common reported adverse effects were of the gastrointestinal type (abdominal pain, diarrhea). In one study, the high prevalence of adverse effects is related to the antimicrobial agents and not to the placebo. No study withdrawal due to adverse events occurred in either of the compared treatment groups in the three trials that provided relevant data.

## 3. Discussion

### 3.1. Main Findings

Here, we demonstrated that the medical device containing xyloglucan, hibiscus and propolis is superior to comparator regimens in terms of clinical effectiveness in adult women with microbiologically confirmed or clinical suspicion of uncomplicated cystitis and is associated with a high patient compliance.

### 3.2. Evidence and Implications for Clinical Practice

In recent years, the use of phytotherapy and nutraceuticals-based medications has gained popularity among non-antibiotic prophylaxis of rUTIs in urological practice [9]. Nowadays, the progressive introduction of phytotherapy and nutraceuticals drugs for the management of symptomatic UTI recurrences represents a reasonable, cost-effective alternative to obviate the massive and reckless use of antibiotics [10]. Xyloglucan, hibiscus and propolis as well as hibiscus and propolis combinations have recently emerged as alternative non-antibiotic approaches in the management of women affected by acute uncomplicated cystitis and rUTIs [5,6,7,8]. The clinical efficacy of XHP on uncomplicated cystitis both in the acute phase and for the prevention of recurrence is due to the following properties:—Mechanical barrier protection (mucoprotectant effect) on intestinal cell mucosa.—Environmental changes in urine (mild decrease of pH) that prevent bacteria proliferation in the urinary tract.

These properties are related to the components in the medical device. Xyloglucan, extracted from the seeds of the tamarind tree (Tamarindus indica), is a ‘mucosal protector’ with the capability to produce a defensive barrier on human epithelial tissues. In vitro studies highlighted how xyloglucan could be efficacious for the management of UTIs by lowering the intestinal reservoirs of uropathogens colonizing the lower intestinal tract, as well as by interfering in the process by which microorganisms contact uroepithelial cell receptors [11,12,13,14].

On the other hand, Hibiscus sabdariffa and propolis have been shown to modify the urine environment, preventing bacteria proliferation and colonization of the urinary tract [15,16].

Several studies have already presented the effects of different mucoprotectant-containing products in women at increased risk for UTIs [5,8]. Herein, we undertook this systematic review and meta-analysis to evaluate the available evidence on medical devices containing a mucoprotectant, hibiscus and propolis, in the management of rUTIs. In this scenario, we selected trials that focused on the efficacy and safety profile of these medications in the prophylaxis of symptomatic recurrences and for improving UTI-related urinary symptoms. Indeed, we included in the selection process all ‘in vitro’ or ‘in vivo’ studies on rat models which evaluated the mechanism of action of mucoprotectants on intestinal epithelial cells [11,12,13,17]. Despite the limited available literature, the results of our study suggest that XHP may represent a promising, non-antibiotic approach to the management of acute UTI and rUTIs. To our knowledge, this is the first meta-analysis to focus on a novel combination of nutraceutical agents as a therapeutical option to manage UTIs. Three RCTs were considered eligible, considering a decrease in symptomatic UTI as the primary endpoint [6,7,8]. The series were comparable in terms of the number of patients considered and the secondary endpoints considered (safety profile or urinary symptoms improvement). Only one study reported the drop-out rate (24%), while only one series required either an active UTI or history of UTI at study entry, although varying criteria and validated tools to define rUTIs were used. Despite the concept of ‘culture-negative’ UTI having emerged, which recognizes the demonstrated effect of antibiotics to resolve UTI symptoms in women who did not demonstrate a culture growth of uropathogens or who had low colony-count infections, it is common in clinical practice to initiate treatment without or before culture results are available, creating a significant bias [18]. In all clinical series, the medical device was adequately tolerated, resulting in excellent compliance, as negligible side effects were registered [19]. Moreover, it showed remarkable results since patients reported a notable clinical improvement of urinary symptoms at 6 months ranging between 19.4–85% [5,6,8].

### 3.3. Study Strengths and Limitations

This meta-analysis is not free from limitations. Firstly, the small observational timeframes and the lack of validated tools for estimating urinary symptoms may represent an intrinsic bias. Furthermore, the restricted number and gender of patients enrolled should be considered a limitation. Moreover, in several series, only patients with an active UTI and unknown history were enrolled and the subsequent UTI was considered a recurrent UTI. Notwithstanding these limitations, our meta-analysis suggests that XHP may represent a potential non-pharmacologic approach for women affected by uncomplicated UTI. Longer-term follow-up and further multicentric series would be necessary to definitively establish the long-term benefits of XHP.

## 4. Materials and Methods

### 4.1. Research Strategy and Literature Search

From September 2020 to April 2021, two independent reviewers (T.C., I.T.) performed the research in PubMed database, Cochrane CENTRAL and Scopus. All disagreements between the two reviewers were resolved by a supervisor (T.B.J.). All references cited in relevant articles were also reviewed and analyzed. The search strategy used was ‘(xyloglucan,) AND (hibiscus) AND (propolis) AND (urinary tract infection OR cystitis)’. As filters, we used: clinical trial, humans, female, English language and adult. Titles and abstracts were used to screen for initial study inclusion. Full-text review was used where abstracts were insufficient to determine if the study met inclusion or exclusion criteria. Two authors independently performed all data abstraction, including evaluation of the study characteristics, risk of bias and outcome measures, with independent verification performed by the senior author. The limited number of studies collected did not require any other authors. The study has been performed in line with the Preferred Reporting Items for Systematic Reviews and Meta-Analyses (PRISMA), the recommendations of the European Association of Urology Guidelines office for conducting systematic reviews and meta-analysis and previous studies [4,20,21]. All selected pre-clinical and clinical trials were used for the systematic review. On the other hand, only clinical trials have been used for the meta-analysis.

### 4.2. Selection Criteria for Study Inclusion in Meta-Analysis and Data Extraction

In this meta-analysis, we included all RCTs involving female patients aged >19 years with microbiologically confirmed or clinical suspicion of uncomplicated cystitis, who were randomized to receive treatment with a medical device containing xyloglucan (or an equivalent mucoprotectant substance), hibiscus and propolis or placebo or other comparator. In line with EAU guidelines, we defined uncomplicated cystitis as acute, sporadic or recurrent cystitis limited to non-pregnant, premenopausal women with no known relevant anatomical and functional abnormalities within the urinary tract or comorbidities [4]. We excluded all male patients and patients with complicated UTIs, defined as reported in the EAU guidelines [4]. We consider for this meta-analysis only studies written in English languages. Abstracts presented at scientific conferences were not considered. Data extraction, risk of bias (RoB) assessment using the Cochrane RoB Tool and quality assessment using the Grading of Recommendations, Assessment, Development and Education (GRADE) [22] approach have been performed by two reviewers (T.C., I.T.) working independently, as described in our previous study [23].

### 4.3. Main Outcomes Measures

The primary endpoint was clinical or microbiological success, defined as the complete (cure) and/or non-complete (improvement) resolution of symptoms at the end of treatment, or microbiological resolutions, in line with the trialists’ definition. Microbiological success (eradication) was defined as the presence of a negative urine culture at the end of treatment [23]. Secondary endpoints included the presence of adverse events, which was defined as any adverse event that was reported at any time during the study period [23].

### 4.4. Statistical Considerations

After extraction, data were pooled to conduct a meta-analysis. Outcomes for continuous variables were expressed as mean difference (MD) with 95% confidence intervals (CI). To analyze dichotomous data, we calculated crude (unadjusted) odds ratios (OR) and log odds ratios. The inverse variance method was used for the combination of the results. Meta-analysis and forest plot diagrams were designed using a random effect model since the sample sizes of the selected studies were different. The ‘trim and fill’ missing study imputation approach was applied to funnel plots, and adjusted overall effect sizes were calculated according to Duval and Tweedie [24]. The software that has been used was the Review Manager (RevMan) Version 5.4 (The Cochrane Collaboration, The Nordic Cochrane Center, Copenhagen 2014). Variations among the studies were calculated using the chi-square test. In addition, I2 was evaluated to indicate the proportion of inconsistency between the selected studies that could not be attributed to chance. Separate analyses regarding all the clinical and microbiological outcomes were performed to compare fosfomycin with each of the different types of antibiotic agent. Finally, the Jadad criteria were used to assess the methodological quality of each of the included trials [25]. The risk of bias was performed by using the Newcastle–Ottawa Scale [26].

## 5. Conclusions

A medical device containing xyloglucan, hibiscus and propolis is superior to comparator regimens in terms of clinical effectiveness in adult women with microbiologically confirmed or clinical suspicion of uncomplicated cystitis and is associated with a high patient compliance. XHP, in this sense, may represent a novel starting grid to expand the armamentarium for uncomplicated UTIs.

## Figures and Tables

**Figure 1 antibiotics-11-00014-f001:**
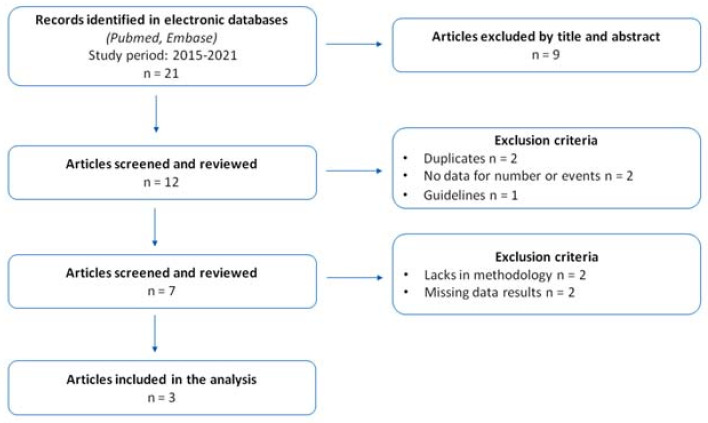
The figure shows the papers’ selection process according to PRISMA recommendations.

**Figure 2 antibiotics-11-00014-f002:**
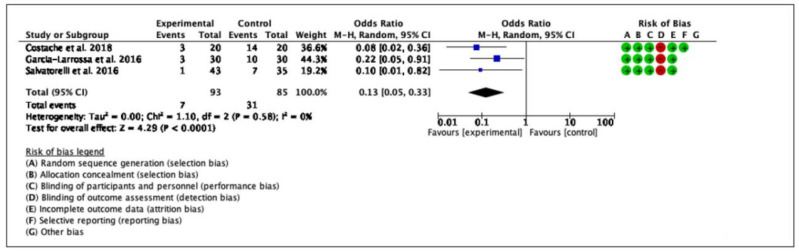
The figure shows a forest plot of the effects of the medical device on women with cystitis in terms of clinical success (improvement/cure).

**Figure 3 antibiotics-11-00014-f003:**
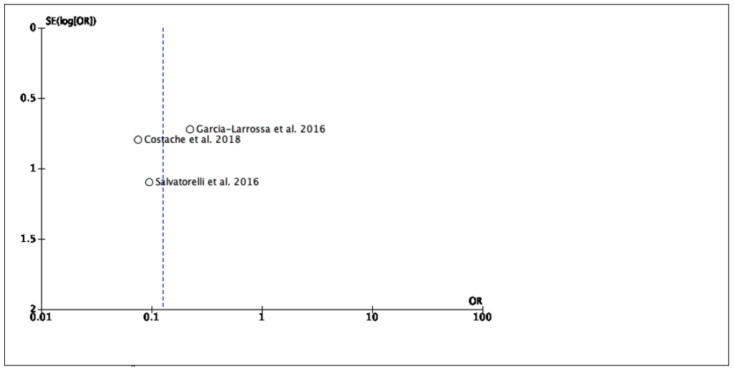
The figure shows the funnel plot for publication bias analysis.

**Figure 4 antibiotics-11-00014-f004:**
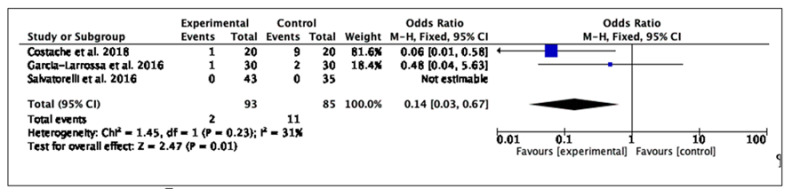
The figure shows a forest plot of the effects of the medical device on women with cystitis in terms of adverse effects.

**Table 1 antibiotics-11-00014-t001:** The table shows all characteristics of all pre-clinical and non-randomized clinical studies. List of abbreviations used in table: MD = medical device; RGHP = reticulated-gelatin hibiscus and propolis; RG = reticulated gelatin (RGHP); SRR = symptomatic recurrence reduction; UTI = urinary tract infection; rUTI = recurrent urinary tract infection; XGHP = xyloglucan–gelose–hibiscus–propolis; AEs = adverse effects; EC = Escherichia coli; XHP = xyloglucan–hibiscus–propolis; XG = xyloglucan–gelose; UT = urinary tract; MPO = myeloperoxidase assay; HSP = H. sabdariffa and propolis.

Variable	De Servi et al., (2016)	Esposito et al., (2020)	Fraile et al., (2020)	Olier et al., (2017)	Cai et al., (2019)
Study design	Evaluation of EC adherence/intracellular invasion in intestinal epithelial cells in vitro	Efficacy of xyloglucan and XG in an in vivo rat model on extraintestinal UTIs	To assess the properties of a medical device containing xyloglucan, propolis and hibiscus to create a bioprotective barrier to avoid the contact of uropathogenic Escherichia coli strains on cell walls in models of intestinal (CacoGoblet) and uroepithelial (RWPE1) cells (derived from normal human prostate epithelium)	Efficacy of RGHP, RG and vehicle on intestinal commensals commonly involved in UTIs in pretreatment animal model (streptomycin)	Single-center, observational, prospective study
Study endpoints	Cytotoxicity, preservation of tight junctions, preservation of paracellular flux, cell invasion, anti-adherence effects	Bacterial growth, intestinal damage and neutrophil infiltration, tight junction permeability, urine volume and pH, effect on bacterial infection of the UT, effect on bladder	Assays of bacterial adhesion, bacterial quantification and antibacterial activity	Bacterial analysis performed in samples recovered from feces	Capability to reduce the number of symptomatic recurrences, efficacy in improving QoL
Study duration	21 days	7 days	-	7–11 days	6 months
Agent	Utipro^®^ *(Xyloglucan, Hibiscus sabdariffa, propolis, silicon dioxide, magnesium* *stearate and corn)*	Xyloglucan and xyloglucan plus gelose (oral)	Xyloglucan (Tamarindus indica) and extracts of HSP	RGHP or RG adult human posology adapted to rat metabolism	Monurelle Plus^®^ (Oral XHP)
Inclusion criteria	Caco-2 cells as intestinal mucosa model	Rats randomly divided into 6 groups — Group 1: control group (vehicle), no EC — Group 2: vehicle + xyloglucan (10 mg/kg daily) — Group 3: vehicle + XG (10 mg/kg daily + 5 mg/kg daily) — Group 4: EC group (109 CFUs/mL) — Group 5: EC + xyloglucan (10 mg/kg daily) — Group 6: EC + XG (10 mg/kg daily + 5 mg/ kg daily)	— Caco-2 cells as intestinal mucosa model — UPEC strains from the Culture Collection (EC expressing type 1 and P. fimbriae (no. 41 from CCUG)). — EC (CCUG), Pseudomonas aeruginosa (CECT111), Staphylococcus aureus (CECT240), Enterococcus faecalis (CECT481)	Control group: female rats with intestinal microbiota not altered by streptomycin (n = 6) Treatment group with RGHP or RG twice daily, highly colonized with EC (n = 6) Study group (n = 30): rats supplemented with streptomycin plus two oral administrations of EC for transient colonization (disrupting colonization cesistance)	Adult women with rUTIs, defined as ≥2 infections in 6 months or ≥3 infections in 1 year
Study protocol/Endpoint	— MTT assay (cytotoxicity) — TEER (tight junctions preservation) — E. coli invasion (inoculation) — Biofilm evaluation	— Infection induced by oral inoculation of 100 μL of E. coli (10^9^ CFUs/mL) administered on days 0, 3 and 6 — Xyloglucan and XG oral treatments administered by oral gavage 2 days before EC oral administration and every day until day 7 — Xyloglucan and XG were at 1, 3 and 10 mg/mL — Feces and urine samples collected from bladder, urethra and intestine, along with urine pH and chemical analysis. — Feces, bladder and urethra were evaluated for bacterial counts f on agar plates. — Intestine and bladder—histological evaluation, myeloperoxidase (MPO) assay and immunostaining of tight junctions	— Adherence assays — Scanning electronic microscopy — Bacterial quantification — Antibacterial activity assay	Fecal samples (days 7–11). EC and other bacteria monitored by selective chromogenic agar plates.	After antibiotic treatment: one capsule a day for 15 days (one cycle every month for 6 months). Clinical and microbiological analyses at baseline (T0) and 1, 3 and 6 months after enrollment. SF-36 validated questionnaire for QoL assessment.
Assessment	Descriptive analysis of quantitative data	Descriptive analysis of quantitative data	Descriptive analysis of quantitative data	Comparison	Descriptive analysis; AEs
Results	— No cytotoxic effect (cell viabilities ≥88%) — Monolayers treated showed higher TEER values — Maintenance of paracellular flux independent of the concentration — Decreased intracellular invasion of EC compared with untreated cells — Decrease in the number of EC cells adhered	-Presence of xyloglucan at all tested concentrations did not significantly affect bacterial growth -Xyloglucan did not induce bacterial lysis in the bacterial strain assessed or cause any irregularities in bacterial cell membranes -Xyloglucan markedly reduced the degree of tissue injury whereas no decrease in E. coli-induced histopathological changes was observed with xyloglucan + gelose -A trend toward decrease in MPO activity was observed in the intestines from rats treated with xyloglucan or XG -XG significantly increased occluding- positive staining-Neither xyloglucan nor XG modified pH values -Bacterial load per gram of urethra (CFU/g) was reduced by 47 and 58% after xyloglucan and XG treatments, respectively -Morphological histological improvement of bladders treated with xyloglucan or XG after EC inoculation	— Maximum concentrations (10 mg/mL for xyloglucan and propolis, 1 mg/mL for hibiscus), did not alter EC integrity — Absence of antimicrobial activity of xyloglucan, propolis and hibiscus — No bactericidal or bacteriostatic activity in any components — Reduction in the number of cells adhered was observed for the three components in comparison with controls (*p* < 0.05) — Highest reduction of adhesion observed with hibiscus at 10 mg/ml	— RGHP significantly reduced the initial median fecal count of EC — EC was no longer detectable in the feces of two of six rats treated with RGHP. — Administration of an equivalent dose of RG alone resulted in a similar decrease of the fecal EC count as compared with initial levels before treatment — RGHP-treated animals showed reduced fecal median counts of Enterococci	Significant reduction in antibiotic use (31.2%) Improvement in urinary symptoms (67.2%)
Conclusions	Utipro^®^ increases the resistance of cell tight junctions and protects cells against the adherence of EC	Barrier effect of xyloglucan on the epithelial intestine reduces the colonization of EC reservoirs, preventing UTI development	— Xyloglucan, propolis and hibiscus are devoid of antibiotic activity — The three components are able to create a physical bioprotective film on intestinal cell mucosa	RGHP reduces fecal colonization by EC and Enterococci or by a dominant human EC strain; RGHP reduces the risk of contamination of the urinary tract by opportunistic bacteria	XHP device improves quality of life in women with rUTIs, reduce recurrences and antibiotic use.

**Table 2 antibiotics-11-00014-t002:** The table shows all characteristics of all included RCTs in the systematic review and meta-analysis.

Variable	Costache et al., (2019)	Garcia-Larrosa et al., (2016)	Salvatorelli et al., (2016)
Study design	Multi-center, randomized, placebo-controlled, parallel group, double blind, phase IV study	Double-blind, placebo-controlled trial	Prospective, randomized, double-blind, placebo-controlled, parallel-group clinical trial
Study endpoints	Frequency of symptomatic rUTI recurrence, AEs	Safety, efficacy of reticulated RGHP in improving urinary symptoms and reducing the need of antibiotics for UTIs	% of symptomatic cystitis recurrences over 6 months, number of micturitions as registered in the diary and dysuria based on a quantitative scale
Study duration	40–90 days	5–11 days	6 months
Agent	XGHP (oral)	RG with hibiscus and propolis	RG Hibiscus sabdariffa calyx extract and propolis.
Inclusion criteria	Adult (men and women) ≥ 18 yrs with uncomplicated UTI	— Adult ≥ 18 years — ≥1 symptoms of UTI < 72 h prior to study entry — Positive dipstick urine test	Adult women ≥ 18 years with acute cystitis symptoms and a ciprofloxacin-susceptible isolate in urine culture
Study protocol/Endpoint	Antimicrobial + 2 daily capsules/ XGHP for 5 days, then 1 capsule of XGHP for 5 days. Follow-up: 1 capsule of XGHP or placebo for 15 consecutive days/month for 2 months	1:1 MD vs placebo for 5 days — Need for antibiotic rescue treatment — Change in symptoms of UTI from baseline to end treatment	Ciprofloxacin 500 mg/day and 2 MD capsules/day vs. ciprofloxacin 500 mg/day and matched placebo for 5 days + two additional cycles of MD for 2 weeks at months 1 and 2 after initial treatment.
Assessment	Physical examination, Likert scale, urine testing and culture (≥10^3^ CFU/mL = positive)	Comparison	Descriptive analysis; AEs
Results	AEs: 5% unrelated to study products; Efficacy analysis: significant reduction of main urinary symptoms (each *p* < 0.004)	— Adverse effects not related to MD — Reduced risk of antibiotic rescue treatment: 33.3% — RGHP superior to placebo for controlling UTI	No recurrence after 1 month (MD-treated) SRR: -19.4% among patients treated reduced by 19.4% Dysuria scores improvement after 20 days of treatment No severe AEs were observed
Conclusions	XGHP as adjuvant therapy to first-line antimicrobials for treatment of uncomplicated UTIs in adults	RGHP reduced the risk of antibiotic treatment and improves UTI symptoms	New MDs prevent the recurrence of uncomplicated cystitis MDs reduce antibiotic use in management of UTIs in women

List of abbreviations used in table: MD = medical device; RGHP = reticulated-gelatin hibiscus and propolis; RG = reticulated gelatin; SRR = Symptomatic recurrence reduction ; UTI = Urinary Tract Infection; rUTI = recurrent urinary tract infection; XGHP = Xyloglucan-gelose-hibiscus-propolis; AEs = Adverse Effects; EC = *Eschericia coli*; XHP = Xyloglucan-Hibiscus-Propolis; XG = Xyloglucan-gelose; UT = Urinary Tract; MPO = Myeloperoxydase Assay; HSP = *H. sabdariffa* and propolis.

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
