# Peer review of "Xyloglucan, Hibiscus and Propolis in the Management of Uncomplicated Lower Urinary Tract Infections: A Systematic Review and Meta-Analysis"

_antibiotics, 2021, doi:10.3390/antibiotics11010014_

Round 1
Reviewer 1 Report
First of all, I want to congratulate the authors for their efforts.
Adjuvant therapy in urinary tract infections is a very largely debated subject. This area of interest represents an important aspect of modern medicine as long as therapists are daily confronted with difficulties in treating this kind of pathology. Adjuvant treatment already gained its place in urinary tract infections treatment and demonstrated its efficacy in delaying recurrences.
I recommended this paper be accepted in the present form.
Author Response
Many thanks for your kind comments.
Reviewer 2 Report
This is a systematic review of several preclinical and clinical papers.Different herbal extracts of different duration (from 5 days to 6 months)
and different components (hibiscus, propolis and sabdariffa cup extract,
etc. versus placebo and ciprofloxacin) were used in clinical studies,
which ultimately makes the study group very heterogeneous.
Furthermore, the number of patients involved is low with short follow-up.
However, I find this study interesting and well-designed
given all the limitations.
Author Response
Many thanks for your comments. We agree with you about the low number of patients included in the SR and the short follow-up period. These considerations have been discussed in the study limitations.
Reviewer 3 Report
The authors conducted a systematic review and meta-analysis of 3 articles and showed that medical devices containing xyloglucan, hibiscus, and propolis could be effective toll in the management of uncomplicated lower urinary tract infections. The study could be valuable for the broad readers: gynecologists, urologists, and public health policy.
I have no significant concerns for the given version of the manuscript.
However, I would like to share with the authors some minor suggestions:
Introduction:
Please specify the prevalence of recurrent uncomplicated urinary tract infections in women, which authors’ assessed as a “high.” Please provide the details about the “impact of those on patient’s quality of life.” The authors will raise readers’ interest in the article with this specific information.
Results:
Figure 1. Please add the additional right box explaining why authors excluded four articled from the seven studies that were “screened and reviewed.”
Author Response
Many thanks for your helpful comments.
- In line with your suggestion, the sentence: "The prevalence of recurrent uncomplicated urinary tract infections (rUTIs) in women is high as well as the impact of those on patients’ quality of life [1-2]", has been changed with the follow: "The prevalence of recurrent uncomplicated urinary tract infections (rUTIs) in women accounts for about 27% and the impact of those on patients’ quality of life is high [1-2]".
- The following sentence has been added to the introduction, in line with your comment: "In particular, suffering from recurrent urinary tract infection has a detrimental influence on patient quality of life, in terms of the level of stress and anxiety [2].
- Figure 1 has been changed in line with your comment.